# Enhancing the Hardened Properties of Recycled Concrete (RC) through Synergistic Incorporation of Fiber Reinforcement and Silica Fume

**DOI:** 10.3390/ma13184112

**Published:** 2020-09-16

**Authors:** Babar Ali, Hawreen Ahmed, Liaqat Ali Qureshi, Rawaz Kurda, Hisham Hafez, Hussein Mohammed, Ali Raza

**Affiliations:** 1Department of Civil Engineering, COMSATS University Islamabad–Sahiwal Campus, Sahiwal 57000, Pakistan; babar.ali@scetwah.edu.pk; 2Department of Highway Engineering Techniques, Erbil Technical Engineering College, Erbil Polytechnic University, Erbil 44008, Kurdistan-Region, Iraq; hawreen.a@gmail.com; 3Scientific Research and Development Center, Nawroz University, Duhok 42001, Kurdistan-Region, Iraq; 4CERIS, Civil Engineering, Architecture and Georresources Department, Instituto Superior Técnico, Universidade de Lisboa, Av. Rovisco Pais, 1049-001 Lisbon, Portugal; 5Department of Civil Engineering, Swedish College of Engineering and Technology, Wah Cantt 47060, Pakistan; liaqat.qureshi@uettaxila.edu.pk; 6Department of Mechanical and Construction Engineering, University of Northumbria, Newcastle upon Tyne NE1 8QH, UK; hisham.hafez@northumbria.ac.uk; 7School of Engineering, University of Edinburgh, Edinburgh EH9 3JL, UK; h.mohammed@zohomail.eu; 8Department of Civil Engineering, Pakistan Institute of Engineering and Technology, Multan 66000, Pakistan; aliraza@piet.edu.pk

**Keywords:** synergistic effects, fiber, recycled aggregate, tensile strength, permeability, silica fume

## Abstract

Portland cement concrete is fragile in tension and it has numerous negative impacts on the environment. To deal with these issues, both fiber reinforcement and recycled materials can be utilized to manufacture sustainable and ductile concrete. In this study, the synergistic effects of high-performance mineral admixture silica fume and glass fiber reinforcement were investigated on the hardened properties of RC. For this purpose, two concrete mix families, namely, NC and RC were prepared. To understand the benefits of synergistic utilization of glass fiber and silica fume, in both NC and RC, 0.5% glass fiber was incorporated with three different levels of silica fume. i.e., 0%, 5%, and 10%. Both strength and permeability-related durability properties were investigated. Results revealed that combined incorporation of 0.5% fiber and 10% silica fume can help in the production of RC having better mechanical and durability performance compared to reference “NC”. Simultaneous incorporation of silica fume and glass fiber produces a combined effect greater than their individual effects on both mechanical and permeability properties of concrete. Silica fume plays a very dominant and positive role in the development of CS, WA, and CIPR of RC, whereas glass fiber plays a vital role in upgrading STS and FS of RC and whereas, with the addition of 0.5% glass fiber, RC can yield 8–9 times higher flexural toughness than that of the plain NC.

## 1. Introduction

Currently, many countries are suffering acute shortages of waste disposal sites to accommodate solid wastes generated during the demolition and construction activities. This issue is severe in many developing countries where rapid urbanization has increased construction and demolition wastes (CDWs) over the past few years. Due to the absence of appropriate recycling conventions in these countries, most of the solid waste is dumped in open landfills that create social and environmental complexities. According to an estimate in 2018 [1], 40 major countries expelled more than 3000 million tons of CDWs into the atmosphere. CDWs can be transformed into recycled aggregates (RA) employing an appropriate crushing method. This practice saves humans from different troubles ensuring eco-friendly disposal of solid wastes and it also relieves natural resources from the distress of constant depletion. Researchers have reported successful applications of RA in different structures, i.e., concrete pavements and buildings [2,3]. Another study [4,5] showed that RA is suitable for steel-reinforced structural applications, i.e., beams and columns.

It is well understood that, despite environmental and economic benefits [6,7], RC has inferior hardened properties compared to natural/normal aggregate concrete (NC). RA containing the low-density mortar is the weakest link in RC. To minimize the deficiencies of RA, it is necessary to strengthen or remove the adhered mortar in RA using a suitable treatment method [8,9]. Another eco-friendly approach is to strengthen the ITZ between the RA and binder matrix by using high-performance mineral admixtures such as silica fume, which also goes by the name of microsilica [10]. RC has been studied with different mineral admixtures such as fly ash, powdered slag, husk ashes, bentonites, and metakaolin [11,12,13,14,15,16]. Mineral admixtures do not only strengthen the ITZ between RA and binder matrix, but their main role is to consume the extra portlandite (CH) produced as a precursor of cement hydration. In this way, strengthening of binder matrix also decreases the strength and durability deficiency of cementitious materials caused by RA.

Another approach to upgrade the mechanical performance of RC is reinforcing it with suitable fiber reinforcements. RC has been investigated with different types of fiber reinforcements mainly to upgrade the ductility and overall mechanical strength of the material. The most commonly investigated type of fiber reinforcement in RC is steel fiber with and without mineral admixtures [13,14,17,18]. Polypropylene, basalt, and glass fibers have also been investigated in RC [19,20,21,22,23,24,25,26]. These studies have shown that fibers mainly contribute to flexural and tensile strength of RC. With a small volumetric dosage of fiber (less than 0.5%), RC shows better tensile and flexural strength than plain “NC” [19,23]. The main contribution of fiber toward strength development of RC is credited to high tensile strength of fiber reinforcement that increases the stiffness and tensile strain capacity of plain concrete. Fibers do not only increase the mechanical performance, but they are also beneficial to some of the durability properties of concrete. They help in effectively controlling the degradation of concrete in the acid attack [27,28] and freeze–thaw [29,30]. They marginally enhance the integrity of concrete owing to their high rigidity and strength, which helps in controlling the degradation of cementitious materials in aggressive environmental conditions. This is also ascribed to marginally increased tensile toughness of concrete due to fiber reinforcement.

The conjunctive use of waste mineral admixtures with fibers has been widely studied on the properties of NC [31,32,33,34,35,36,37,38,39,40,41,42]. Coupling benefits of fibers and fly ash helps in lessening WA capacity of fibrous concretes, improvement in workability, etc. [31,43]. High-performance mineral admixtures like silica fume can improve the dispersion of fibers in the concrete matrix, which can upgrade the contribution potential of fibers toward the strength of NC [38,44].

There are fewer studies [13,18,37,40] that have explored the properties of RC utilizing a combination of both fiber reinforcement and mineral admixture. Xie et al. [18] described that synergistic use of silica fume and fiber can improve the bond strength and dispersion of steel fiber in the concrete matrix, which results in an upgradation of CS and FS of RC. They found that simultaneous incorporation of 8% silica fume with steel fiber produces RC having CS closer to that of the reference “NC”. Another study [45] showed that synergistic incorporation of polypropylene fiber and silica fume can increase the replacement level of RA with NA without losing significant mechanical strength. To the best information of authors, no study investigates the properties of RC containing glass fiber and silica fume. Glass fiber is a cheap and eco-friendlier [46] fiber reinforcement compared to conventional steel fiber [3]. It is corrosion resistant unlike steel; therefore, it ensures the stability of structures for longer duration in aggressive environments. Furthermore, glass fiber is three times lighter than steel fiber. To fulfill the rising demands for sustainable, ductile, and durable cement-based composites, it is necessary to choose those materials (recycled aggregates, waste mineral admixtures, low carbon footprint fibers) that are eco-friendly and durable compared to conventional materials (natural aggregate, Portland cement, and high carbon footprint fibers).

Therefore, the main objective of this research was to examine the individual and interactive effects of glass fiber and silica fume on various hardened properties of concrete. This study also intended for the production of RC integrated with benefits of both fiber and mineral admixtures, which yields better performance than conventional “NC”. For this purpose, two concrete families, namely, NC and RC, were produced with and without glass fiber. Both plain and fiber-reinforced NC and RC mixes were studied with three different levels of silica fume, i.e., 0%, 5%, and 10%. Mechanical performance was evaluated based on the outcomes of CS, STS, and FS testing. Load vs. midspan deflection data of all mixes were also investigated. Permeability resistance of mixes was assessed based on the results of WA and CIP testing. Correlations between studied experimental properties were also analyzed and discussed. The key finding of this research showed that the simultaneous use of silica fume and glass fiber can produce RC having better strength and permeability resistance than “NC”.

## 2. Experimental Methodology

### 2.1. Materials

In this section materials used for the manufacturing of mixes are explained. Portland cement (Bestway, Haripur, Pakistan), classified as Type I according to ASTM C150 cement, was used as the primary binding material [47]. Its main properties are presented in Table 1. Silica fume (Sika, Rawalpindi, Pakistan) was used as a partial replacement of cement containing 90–94% pure microsilica having a specific surface area of 27,000 m^2^/kg. For fine aggregate, Lawrancepur-based quarry sand, dominantly a siliceous sand (Lawrancepure, Attock, Pakistan), was used. Its properties are given in Table 2. For NA, Margalla Hills-based (Taxila, Pakistan) crushed stone was used to manufacture NC. Concrete specimens, having cubic compressive strength 30–35 MPa, were crushed and graded to produce coarse recycled aggregate (RA). The maximum particle size of both NA and RA was 12.5 mm. The main properties of these aggregates are presented in Table 2. Size distribution of aggregate particles in fine aggregate, NA, and RA is illustrated in Figure 1.

For fiber reinforcement, Cem-fill, alkali-resistant-glass, chopped strands were used. The tensile strength and specific gravity of these fibers were 1700 MPa and 2.63, respectively. All the important characteristics of this fiber are given in Table 3. To control the loss in workability of NC and RC mixes due to the incorporation of silica fume and/or fiber, Viscocrete-3130 (Sika Pvt Ltd., Rawalpindi, Pakistan) was used as a water-reducing admixture.

### 2.2. Details of Concrete Mixes and Mixing Method

Two mix “families”, namely, NC and RC, were produced using NA and RA as coarse aggregates, respectively. To study the “synergistic or interactive” effects of glass fiber and silica fume in each of NC and RC family, 0% and 0.5% glass fiber was used with 0%, 5%, and 10% silica fume. Glass fiber dose of 0.5% was measured in volume fraction of concrete that was 13 kg/m^3^ (i.e., 0.5% of 2600 kg/m^3^). Silica fume was incorporated as by-weight replacement of cement. Water-to-binder ratio for all NC and RC mixes was kept constant at 0.38. Full details about composition of mixes are provided in Table 4. To compensate for the loss in effective water and workability, surplus water equal to 80 kg/m^3^ was used in RC families. Coarse RA was air-dried when used in the preparation of RC; therefore, surplus water was compulsory to fulfill the absorption of RA. This was necessary to maintain the amount of effective water in concrete matrix that is required for cement hydration. Viscocrete-3130 was used to maintain the workability.

Concrete mixes were blended in an adjustable speed (rpm) mixer of 0.15 m^3^ capacity. Firstly, aggregates were blended with half of the total water for 4 min at 40 rpm. After that, binding material, fibers, plasticizer, and half of water were charged into the mixer and blended for 6 min at 60 rpm. Subsequently, slump test was performed on all mixes to check for the desired workability (slump of 8–11 cm). After checking for the intended workability, mixer continued to run at a slower speed of 20 rpm until the casting of specimens finished.

### 2.3. Preparation of Specimens for Strength and Permeability Tests

Three main mechanical properties, namely, CS, STS, and FS, were studied to investigate the interactive effects of silica fume and glass fiber on the overall strength performance of NC and RC. Cylindrical specimens of 100 mm diameter × 200 mm length were used for the evaluation of CS and STS. CS was measured according to ASTM C39 [54]. STS was measured as per ASTM C496 [55]. To evaluate load vs. deflection data, test was conducted on 100 mm (width) × 100 mm (height) × 350 mm (length) prismatic specimen under third-point loading according to ASTM C1609 [56]. Peak load in bending test was used to calculate FS. Each reported strength parameter in this research is the average result of three replicate specimens.

To understand the interactive effects of silica fume and glass fiber on permeability resistance of RC and NC, WA and CIP tests were conducted. WA testing was performed on the concrete specimens of 50 mm (height) × 100 mm (diameter) according to ASTM C948 [57]. CIP test was executed on specimens of 100 mm (diameter) × 100 mm (height), as explained by authors [58,59]. Cylindrical specimens for CIP testing were first cured for 28 days in tap water. Cured specimens were then dried in air for 7 days. Then, six air-dried specimens of each mix were dipped in 5% sodium-chloride solution. Three specimens of each mix were tested after 56 days of immersion and the remaining three were tested after 90 days of immersion period. Chloride conditions’ specimens were split into two halves, then the split surface was sprinkled with 0.1 normality AgNO_3_ solution. When nitrates from spraying solution reacted with chloride ions, a visible silver color showed the extent of CIP into the specimens. Observation and calculation of CIP was done as presented in Figure 2.

## 3. Results and Discussion

### 3.1. Compressive Strength (CS)

The results of CS testing are displayed in Figure 3. The relative analysis of CS results is also provided in Figure 3b. The trends in the results show that CS of both NC and RC experienced huge upgradation with the combined addition of silica fume and glass fiber. RC, with the help of both silica fume and glass fiber, can show higher CS than reference “NC” mix. RC showed 15% less CS than NC. Lower strength of RC than NC can be blamed on the existence of many interfacial transition zones (ITZs) in RC and lower density of recycled aggregates than natural ones [11,60,61]. Namely, four types of ITZs exist in RC: (1) ITZ between mineral aggregate of RA and adhered mortar, (2) ITZ between mineral aggregate of RA and new mortar, (3) ITZ between NA and new mortar, and (4) ITZ between new mortar and adhered mortar of RA. The presence of too many ITZs in RC may lead to premature bonding failure between aggregates and binder matrix under compressive loads [62]. Moreover, RC showed less stiffness and large lateral deformation under compressive load than NC [40]. The significant loss in CS with 100% replacement of coarse NA with RA reported in different studies is shown in Figure 4. These findings showed no clear trend about how much loss in CS is expected with the replacement of coarse NA with RA. The average of CS loss reported in 11 different studies was about 17%, which is closer to CS loss experienced in the present study. High CS loss due to full replacement of coarse NA with RA was observed in the low strength classes (25 MPa) of concrete [9], whereas high strength classes did not show significant CS loss at full replacement of coarse NA with RA, i.e., 7–8.5% [11,14]. So far, to the authors’ understanding, high CS loss is observed in the low strength classes due to a very high volume of coarse aggregate (high aggregate to binder ratio), whereas in high strength classes, like in the present, lesser coarse aggregate volume was used. Therefore, the negative effect of full replacement of NA with RA on CS was minimized due to low aggregate-to-binder ratio.

The addition of silica fume substantially advanced the CS of RC. The 5% and 10% incorporation of silica fume improved the CS of RC by 11.1% and 19%, respectively, whereas NC experienced smaller CS increments compared to RC, i.e., 7.7% and 11% at 5% and 10% silica fume, respectively. This is due to the different pozzolanicity potentials of RC and NC, as RC contains CH in the adhered mortar. Therefore, pozzolanicity potential of RC is higher than NC. The strengthening of ITZs between old mortar and new mortar due to pozzolanic reaction between silica fume and CH contributed to an additional net gain in the strength of RC. The CS of RC at 10% silica fume surpassed that of the reference “NC”. Xie et al. [40] also reported noticeable strength increments (of about 17%) due to 8% silica fume addition in RC.

The use of glass fiber contributed 2–4% increment to the CS of RC and NC. These improvements are mainly credited to the increased integrity of concrete matrix [14]. Moreover, fibers do not play a very positive role in the development of CS and they are more useful in the tensile strength and flexural strength [10,63]. In the synergistic effect of fiber and silica fume on CS, the role of silica fume is very significant as it causes the major upgradation in the CS. Fiber-reinforced RC with 5% and 10% silica fume outperformed reference NC mix by margins of 0.8% and 5.9%, respectively, whereas RC-GF-SF5 and RC-GF-SF10 showed 18% and 25.7% higher CS than plain “RC”. Silica fume improved the density of particle packing in binder matrix and contributed to growth microstructure (more CSH gel), whereas glass fiber prevented the premature failure of RC due to increased integrity of concrete matrix. Silica fume also contributed toward the bond strength of glass fiber. As the density of binder matrix improved, the interfacial area between fiber and matrix increased and that advanced the bond strength of fiber [64,65]. Due to improvement in bond strength of fiber, the net gain in CS due to fiber addition increased with the addition of silica fume. For example, at 0%, 5%, and 10% silica fume, glass fiber showed 5%, 8%, and 7% improvement in CS.

### 3.2. Splitting Tensile Strength (STS)

Variation in STS of NC and RC with the incorporation of silica fume and glass fiber is shown in Figure 5. Relative analysis of results is also presented in Figure 5a. The results show the positive synergistic effects of fiber and fume on the STS of both NC and RC.

As expected, RC showed 9% lower STS than NC. This deficiency inherits from the weak bond between adhered mortar and new binder, due to high water absorption capacity of RA in RC. Although high angularity of RA compared to NA compensates some loss in STS, still RC underperforms marginally as compared to NC. The STS loss reported in this study was compared with that reported in 11 different studies, see Figure 6. In seven studies, the loss due to 100% replacement of NA with RA was lower than 13.3% and in the remaining four, loss was higher than 13.3%. Highest STS loss was observed in those studies [9,66,70] investigating mixes with high aggregate-to-binder ratio, whereas, like the present study, low strength loss was observed in studies [13,14,67,68,69] having low aggregate-to-binder ratio.

STS of RC experienced a net increase of 7% and 14% at 5% and 10% incorporation of silica fume. These improvements were already ascribed to strengthening of binder matrix due to silica fume. Unlike CS, STS of both NC and RC experienced less improvement due to addition of silica fume. This is because the filling effect of silica fume particles contributes more to the compression stiffness of concrete, but it is not efficient in increasing tensile strength. Unlike NC, RC undergoes more improvement in STS due to silica fume addition. Silica fume increased the tensile capacity of concrete matrix by improving the bond between RA and binder matrix. Mainly, strengthening of binder matrix due to the potential pozzolanic reactions between microsilica and CH helped in tensile capacity building of both NC and RC. RC owing to high CH content offers more potential for STS development with silica fume than NC. This is well established in literature that RC is befitted more than NC due to addition of pozzolanic materials than NC [11,62,72]. Owing to developments in microstructure, RC with 10% silica fume performed better than reference NC.

Both RC and NC experienced a marginal improvement in STS due to glass fiber addition. With 0.5% fiber incorporation, STS of NC and RC increased by 17% and 21%, respectively. Fiber-reinforced RC showed 10% higher STS than plain “NC”. These improvements due to fibers are attributed to increased crack-bridging capability of both RC and NC. Deficiency or brittleness of concrete matrix in tension is overcome by intrusion of high tensile strength fiber material.

Synergistic effect of fiber and silica fume further enhanced the tensile capacity of RC. Combinations of 5% silica fume +0.5% fiber and 10% silica fume +0.5% fiber improved the tensile strength of RC by 23 and 29% with respect to reference “NC”, respectively. Incorporation of silica fume with fiber does not only combine the benefits of mineral admixture and reinforcement, but it also helps in advancing the efficiency of fibers to enhance the strength properties of concrete. For example, without silica fume, net increment in STS of RC due to fibers was about 21%, but after the addition of 5% and 10% silica fume in binder, net increment in STS due to fiber was increased to 26–27%. This is the clear indication of augmentation in bond strength of fibers with concrete matrix. Therefore, the quantity of benefits due to combined incorporation of fume and fiber was noticeably higher than the sum of benefits achieved due to separate incorporation of fiber and silica fume. These results confirmed the synergizing potential of fiber reinforcement and silica fume in both NC and RC.

### 3.3. Flexural Behavior (Load Versus Midspan Deflection)

Data of load versus midspan deflection of prismatic specimens was recorded as per ASTM C1609. This data are plotted in Figure 7. Area under the curve of load-deflection data was used to calculate flexural toughness of mixes. Flexural toughness of each mix is shown in Figure 8. The peak loads were determined and used in the evaluation of flexural strength (FS) (see Figure 9).

Glass fiber marginally influence the peak load and postpeak behavior of both RC and NC. Descending curves for glass fiber-reinforced mixes were marginally flatter than those of the plain mixes. This showed marginal improvement in tensile toughness and residual strength of concrete due to the addition of glass fiber. Fibers can carry noticeable amount of load when the plain matrix of concrete starts failing after the peak load; therefore, fibers ensure a ductile failure by preventing the complete rupture of specimen. Furthermore, silica fume increased the peak load and flexural stiffness, mainly because of the improvement in the microstructure of binder matrix but it did not modify the postpeak behavior of RC and NC mixes.

To measure the flexural toughness, area under the load deflection curve was calculated. Flexural toughness of each mix is shown in Figure 8. Flexural toughness of NC and RC was improved by more than 9 times due to addition of glass fiber. Tremendous increment in flexural toughness was ascribed to increased displacement under the flexural loading. Moreover, there was no significant difference between toughness values of NC and RC. This might be because RC mixes, despite withstanding lower peak loads, retained relatively high residual strength after the peak load compared to plain NC mixes, as shown in Figure 7.

### 3.4. Flexural Strength

The change in FS of both NC and RC with the changing silica fume content and glass fiber is shown in Figure 9. As anticipated, RC showed lesser FS than NC, mainly because of presence of weak coarse RA. Full replacement of NA with RA did not damage FS and splitting tensile strength (STS) like it did compressive strength (CS). Despite low density, the angularity of RA might facilitate the internal aggregate locking and friction that compensate for some loss in both STS and FS due to replacement of NA with RA [58,70,73]. The positive effects of silica fume and glass fiber on FS are like those observed in the results of STS testing. Like STS, FS of both NC and RC also underwent substantial improvements, of 23–25%, due to glass fiber reinforcement. It was evident from the previous studies that both STS and FS were more benefited from fibers compared to CS [74,75]. With separate incorporation of 0.5% glass fiber and 10% silica fume, RC showed 3 and 13% higher FS than reference “NC”.

Coupling fiber and silica fume upgraded the FS of RC by more than 40%. Glass fiber-reinforced RC at 5% and 10% silica fume contents showed 39% and 46% higher FS than plain RC, respectively. Due to synergistic effect, silica fume improved the net efficiency of fibers by 15–20%. It is because the strengthening of binder matrix increased grip of binder over fibers [64,65], which helped in efficient transfer of tensile stress from concrete matrix to high tensile strength fiber.

The correlation of FS with CS and STS is shown in Figure 10. These correlations were drawn without considering the effects of silica fume and fiber reinforcement on strength properties. FS showed a weaker and stronger correlation with CS (R^2^ = 0.5) and STS (R^2^ = 0.98), respectively. This is because of different behaviors of CS and STS with the inclusion of fibers in both NC and RC. CS was strongly dependent on (1) density and (2) growth of microstructure in the concrete matrix, whereas fiber addition did not change the density and microstructure of concrete significantly. Therefore, CS was largely affected by silica fume addition. On the other hand, STS and FS, both tensile strength parameters, depend on the (1) density, (2) microstructural developments, and (3) crack-bridging capability of the material. The simultaneous addition of both silica fume and fiber contributed to all these three parameters. Therefore, both STS and FS showed a similar pattern in the variation of results with fiber and silica fume addition. A strong linear correlation between these two tensile strength parameters indicates that one parameter can be accurately estimated from the other. FS, also known as modulus of rupture, is widely used in determining the design thickness of concrete pavements. Measuring FS of plain and fibrous concretes is a very critical task and requires good quality control while preparing and testing of prismatic specimens. Therefore, STS can be used for the accurate estimation of FS of fiber-reinforced composites for the thickness design of highway pavement.

### 3.5. Water Absorption (WA)

WA is an implicit estimate of durability of cement-based composites. It represents the percent of water-permeable voids in concrete and both strength properties and permeability-resistance against chemicals largely depend on the voids’ ratio of concrete. Influence of glass fiber with varying silica fume content on WA capacity of both NC and RC is shown in Figure 11.

WA capacity of concrete increased by 22% when coarse NA was completely replaced by RA. High absorption capacity of RC is usually ascribed to the presence of porous mortar in coarse RA [76]. Furthermore, high water demand of RC to satisfy the WA of coarse RA increases the pore volume of concrete. In the cases of both NC and RC, a small increase of 3–4% was noticed in WA with the inclusion of glass fiber. Increase in WA due to fibers was mostly blamed on the increase in connectivity of pore volume. Small microchannels created by the weak ITZ between fiber and concrete matrix may facilitate the penetration of water into NC and RC specimens. Moreover, bundle nature and poor dispersion of glass fibers in concrete matrix may also favor sorption of water.

WA of RC and NC substantially reduced with the addition of silica fume into the binder. It is because microparticles of silica fume reduced the pore volume of concrete by fitting between the aggregates and cement particles. This led to the improvement in the density of microstructure. Connectivity between the pores was also significantly reduced, due to the development of CSH as a result of pozzolanic reaction. Similar developments happened in the binder matrices of both NC and RC due to silica fume addition. Since pozzolanic reactions are also possible at the ITZs between RA and binder matrix, therefore, RC underwent huge improvements in concrete matrix compared to NC due to incorporation of silica fume. For example, NC experienced drops of 16% and 26% in WA at 5% and 10% silica fume addition, respectively, whereas WA of RC dropped by 20% and 31% at 5% and 10% silica fume, respectively. RC with 5% and 10% silica fume showed better WA resistance than reference “NC”.

Synergizing silica fume and fiber minimized the negative effect of fibers on WA capacity of RC and NC. Unlike pure cement-based mixes, silica fume-added mixes showed no noticeable difference between the WA capacities of plain and fiber-reinforced mixes, see Figure 11a. This is because smaller particles of silica effectively block/fill the interconnected pores along the fibers and, moreover, growth of CSH gel around fibers also reduce the chances of water permeability along fibers. The negative effect of bundled nature of glass fiber may also be reduced as the microparticles of silica fume fit into the spaces between fiber filaments. On the other hand, in synergistic effect of fiber and silica fume, only silica fume played a positive and very dominant role. These results show that to manufacture ductile composites, sometimes, permeability resistance may be jeopardized. Therefore, to minimize the negative impacts of fibers on composites, silica fume or other mineral admixtures can play a useful role.

### 3.6. Chloride Ion Penetration (CIP)

CIPR of cement-based materials ensures the protection of steel rebars against corrosion in concrete structures and, hence, it is a key durability parameter. In this research, chloride ion penetration (CIP) refers to the depth (as shown in Figure 2) up to which chlorides from 5% NaCl solution penetrate the matrix of concrete. Chloride ion penetration (CIP) values of specimens exposed to 5% NaCl solution for 56 and 90 days are shown in Figure 12. Net increase or decrease in CIPR of concrete mixes with the addition of silica fume and glass fiber is illustrated in Figure 13.

CIPR of RC was 28–31% lower than that of the NC. This is because of the higher absorption capacity of RC compared to NC. A net loss of 20% and 16% in CIPR due to 100% replacement of NA with RA was also observed in the studies of Poon et al. [62] and Koushkbaghi et al. [13], respectively. On the other hand, fiber-reinforced NC and RC showed 3–6% lesser CIPR than their corresponding plain mixes. Reduction in CIPR due to fiber addition can also be linked to the increased connectivity of pores along fiber filaments [72].

Silica fume was very beneficial to CIPR of NC and RC. NC with 5% and 10% silica fume content showed 19–23% and 53–59% higher CIPR than reference “NC”, respectively. RC incorporating 10% silica fume showed 13–19% higher CIPR than reference “NC”. This shows that silica fume can overcome the negative effects of RA on permeability resistance of concrete. Furthermore, synergistic effects of fiber and silica fume minimize the negative effect of fiber reinforcement on CIPR. Owing to silica fume, RC-GF showed 10–15% higher CIPR than reference “NC” mix.

Since both WA and CIP are dependent on the density and microstructural development of concrete matrix, there was a robust connection between these two parameters, as shown in Figure 14. Measuring CIP is much more difficult than WA; therefore, such correlations offer a useful tool for determination of CIPR of concrete mixes by simply evaluating the WA. On the other hand, compressive strength is also dependent on the density and microstructural growth (i.e., CSH formation) in the concrete matrix. Indirectly, compressive strength can be related to durability indicators such as WA and CIP. In Figure 15, the relationship among these three parameters is shown. As compressive strength is advanced due to improvements in concrete matrix, both WA and CIP show a declining trend.

## 4. Conclusions

An experimental campaign was executed to investigate the synergistic effects of silica fume and glass fiber on hardened properties of NC and RC. Following are the key conclusions drawn from this study:(1)Simultaneous incorporation of silica fume and glass fiber provides excellent synergistic effects on strength and permeability resistance of both RC and NC. Silica fume incorporation improves the bond strength of glass fiber with binder matrix and it can also help in better dispersion of glass fiber.(2)The separate incorporation of 10% silica fume and 0.5% glass fiber improved the CS of RC by 19% and 4.5%, respectively, whereas combined effect of silica fume and 0.5% glass fiber improved the CS by 25.7%. RC with 5–10% silica fume and 0.5% glass fiber showed higher CS than reference “NC”.(3)Synergistic effect of fiber and silica fume was very prominent in the results of STS and FS. The 10% silica fume and 0.5% glass fiber separately enhanced the STS of RC by 14% and 21%, respectively but the combined effect of silica fume and fiber addition enhanced the STS of RC by 41.3%. This indicates an increase of 25% in the efficiency of glass fiber in STS. RC with 10% silica fume and 0.5% fiber outperformed reference “NC” by a margin of 29%. FS testing showed similar trends in results.(4)FS of fiber-reinforced NC or RC can be fairly estimated from the STS without considering the effect of supplementary material (fiber or silica fume), whereas FS showed poor correlation with CS for fibrous mixes. Flexural toughness of RC can be improved by more than 8–9 times by inclusion of 0.5% glass fiber compared to plain NC.(5)WA and CIP increased with the incorporation of RA and fiber into concrete. The difference between WA and CIP values of plain and fiber-reinforced concretes (both NC and RC) reduced with the inclusion of silica fume into binder. Silica fume effectively controlled the loss in permeability resistance due to fiber reinforcement in the cases of both NC and RC. Plain and glass fiber-reinforced RCs with 10% silica fume showed superior WA and CIP resistance than reference “NC”.

## Figures and Tables

**Figure 1 materials-13-04112-f001:**
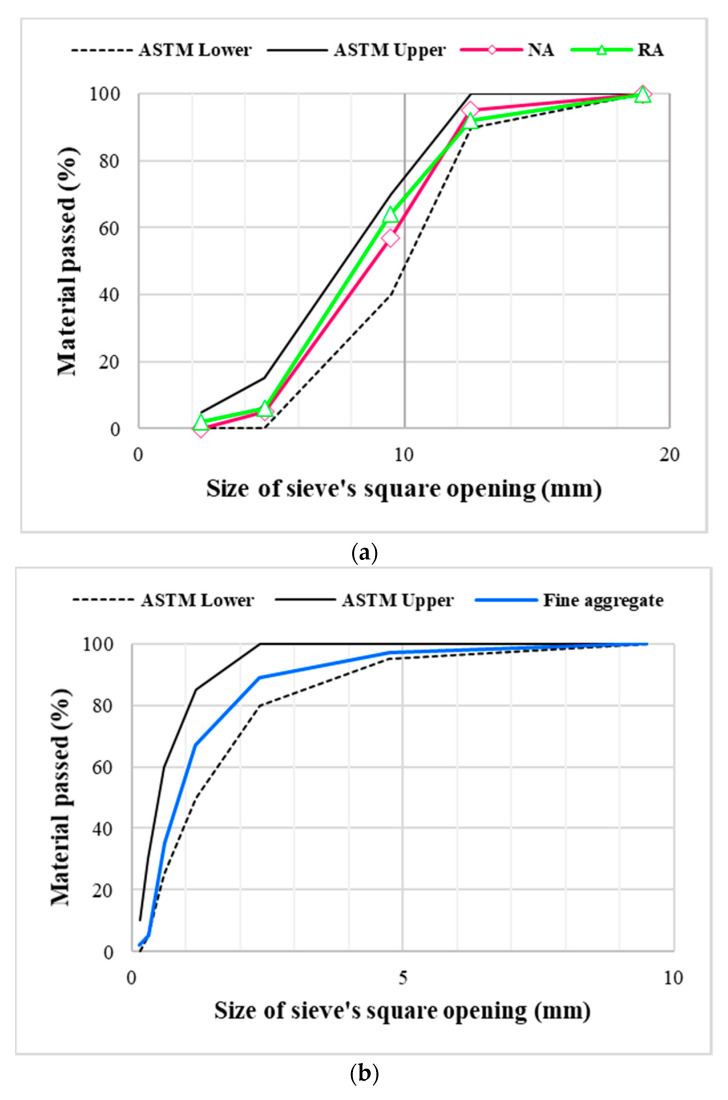
Aggregates’ gradation. (**a**) Coarse aggregate, (**b**) fine aggregate.

**Figure 2 materials-13-04112-f002:**
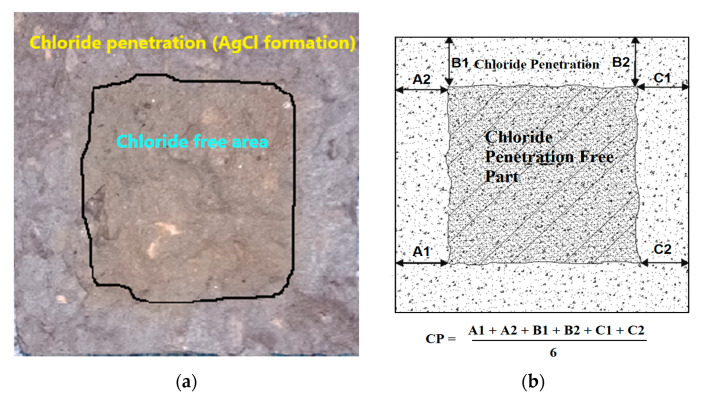
(**a**) Observation and (**b**) calculation of chloride ion penetration (*CIP*).

**Figure 3 materials-13-04112-f003:**
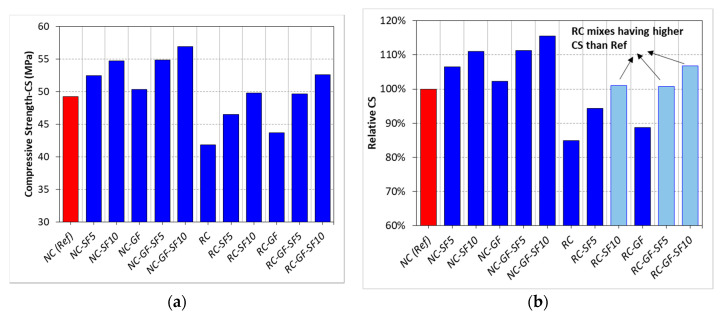
(**a**) The results *CS* of each mix, (**b**) relative analysis of *CS.*

**Figure 4 materials-13-04112-f004:**
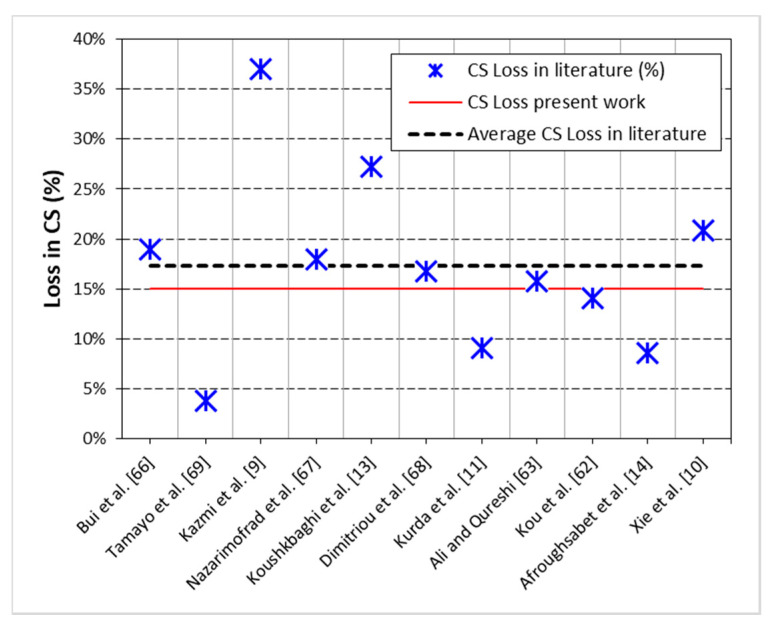
Trends in *CS* loss due to 100% replacement of coarse NA with RA independent of strength class of concrete (20–85 MPa) reported in the past studies (data from [9,10,11,13,14,62,63,66,67,68,69]).

**Figure 5 materials-13-04112-f005:**
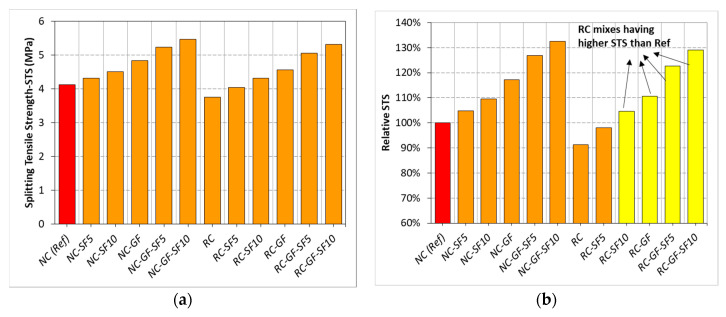
(**a**) Results of *STS* testing of each mix, (**b**) relative analysis of *STS.*

**Figure 6 materials-13-04112-f006:**
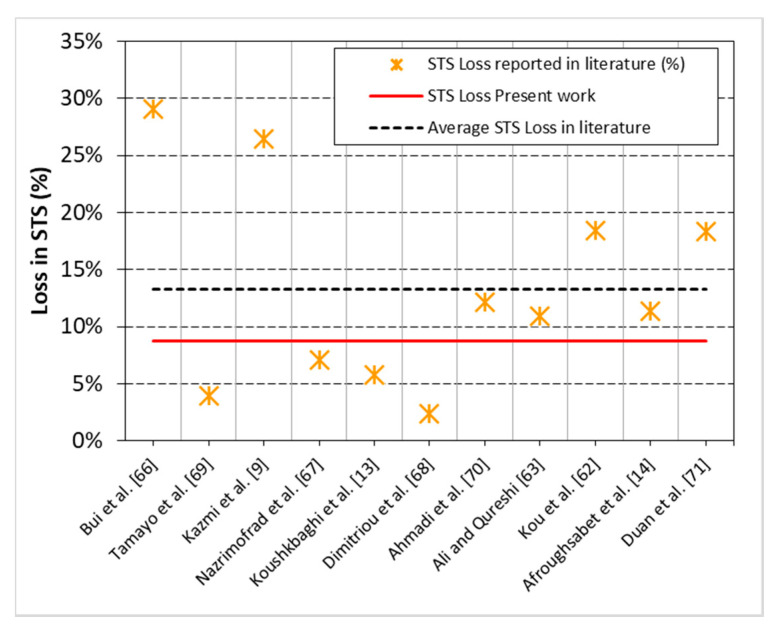
Trends in *STS* loss due to 100% replacement of coarse NA with RA independent of strength class of concrete (20–85 MPa) reported in the past studies (data from [9,13,14,62,63,66,67,68,69,70,71]).

**Figure 7 materials-13-04112-f007:**
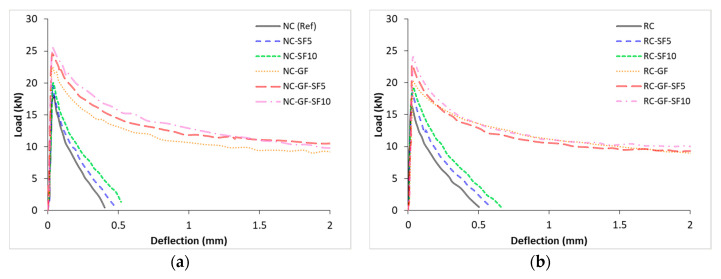
Load vs. midspan deflection of plain and glass fiber-reinforced (**a**) NC and (**b**) RC.

**Figure 8 materials-13-04112-f008:**
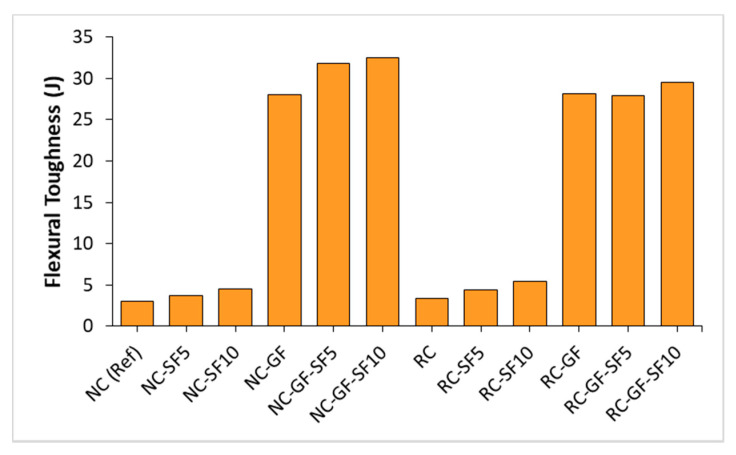
Flexural toughness (kN-mm/Joules) of NC and RC mixes.

**Figure 9 materials-13-04112-f009:**
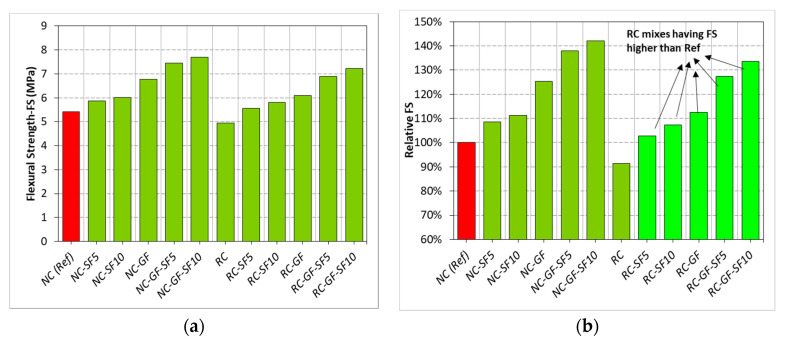
(**a**) The results of *FS* testing of each mix, (**b**) relative analysis of *FS.*

**Figure 10 materials-13-04112-f010:**
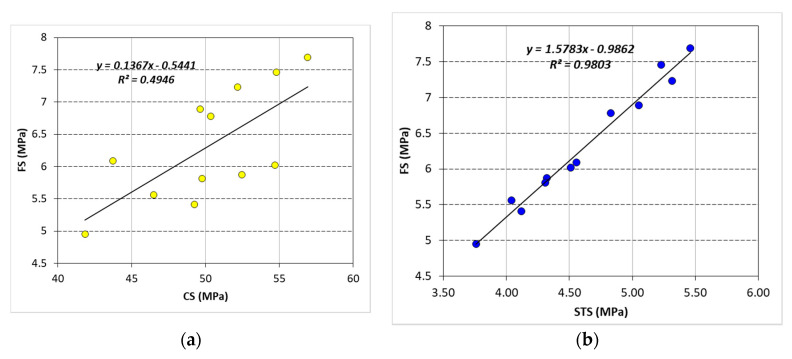
(**a**) Linear correlation between CS and *FS,* (**b**) linear correlation between *STS* and *FS.*

**Figure 11 materials-13-04112-f011:**
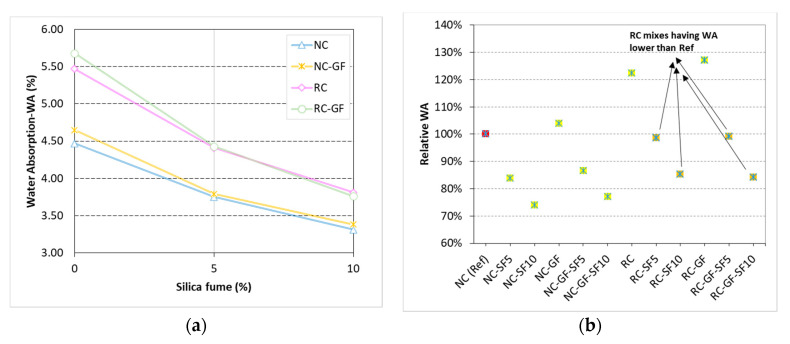
(**a**) Variation in *WA* capacity of NC, NC-GF, RC, and RC-GF with the varying content of silica fume, (**b**) relative *WA* capacity of each mix with respect to reference “NC”.

**Figure 12 materials-13-04112-f012:**
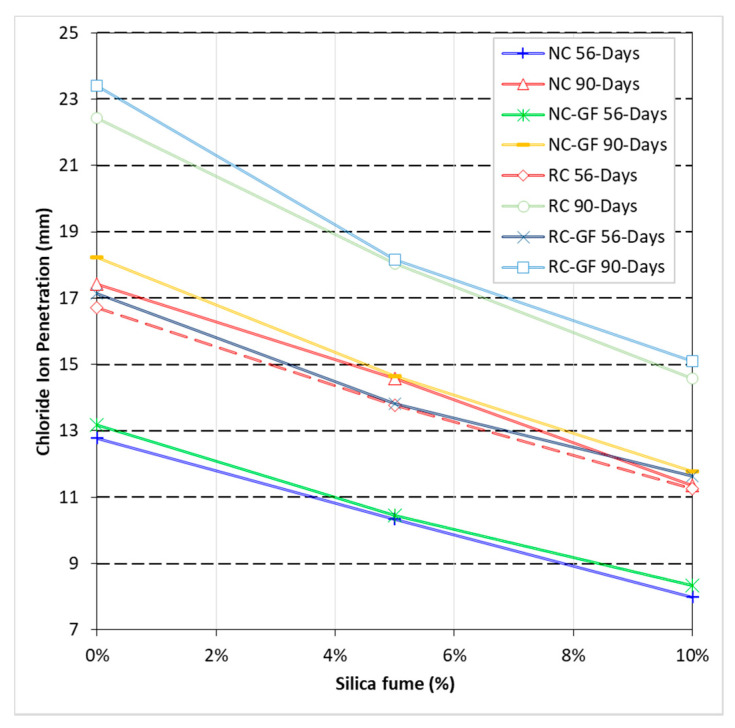
Effect of silica fume and glass fiber on *CIP* value of NC and RC.

**Figure 13 materials-13-04112-f013:**
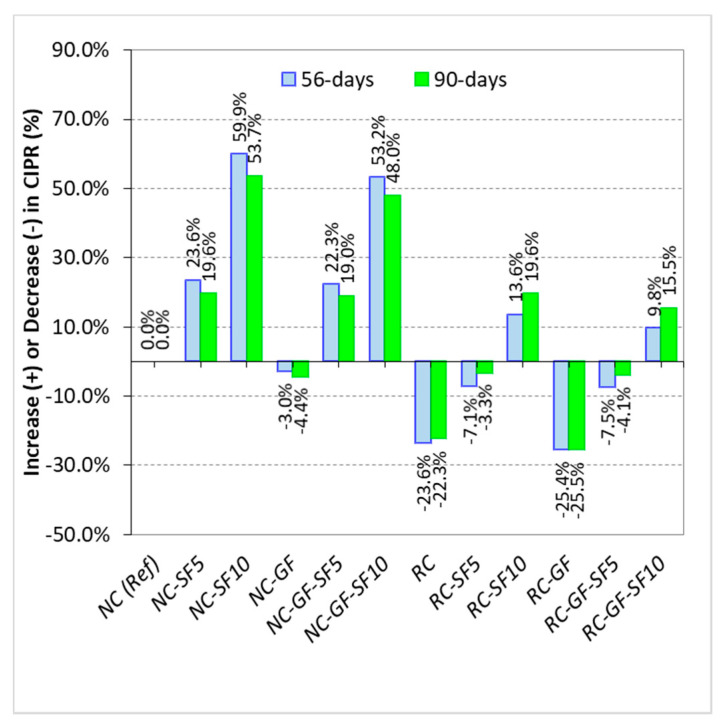
Change in *CIPR* of NC and RC with the variation in silica fume and glass fiber content.

**Figure 14 materials-13-04112-f014:**
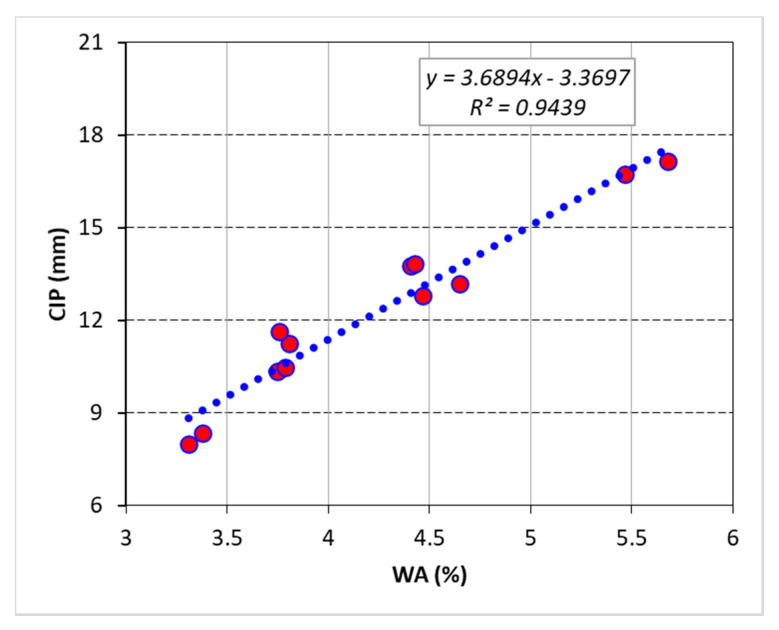
Linear correlation between *CIP* and *WA.*

**Figure 15 materials-13-04112-f015:**
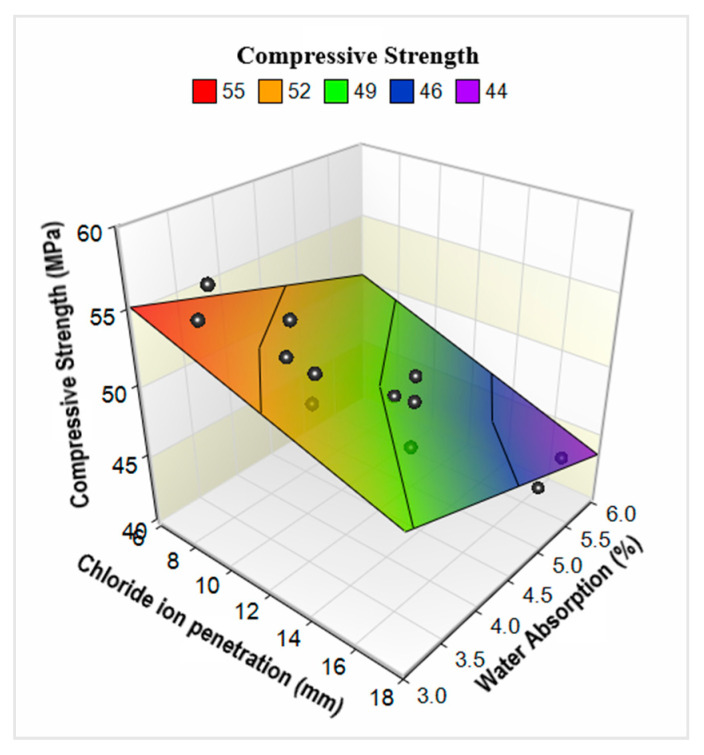
Correlation between compressive strength-(CS), chloride ion penetration-(*CIP*), and water absorption (*WA*) capacity.

**Table 1 materials-13-04112-t001:** Properties of binders.

Composition	Cement	Silica Fume	Physical Properties	Cement	Silica Fume
Silica (SiO_2_)	23.4%	90–94%	Specific-gravity [48]	3.10	2.25
Alumin (Al_2_O_3_)	5.7%	1.12%	Specific-surface-area (m^2^/kg) [49]	324	27,000
Iron-oxide (Fe_2_O_3_)	4.1%	0.07%	Consistency (%) [50]	29.65	-
Lime (CaO)	64.45%	0.67%	Initial-setting time (mins) [51]	115	-
Magnesia (MgO)	2.2%	0.01%	Final-setting time (mins) [51]	234	-
Sulfur-trixoxide-(SO_3_)	2.8%	-	Soundness [52]	No expansion	-
Sodium oxide-(Na_2_O)	0.4%	-	Compressive strength (28-days) [53]	47.56	-
Potassium oxide-(K_2_O)	0.5%	-	-	-	-
Loss on ignition (%) (LOI)	0.61%	1.37%	-	-	-

**Table 2 materials-13-04112-t002:** Properties of aggregates.

Characteristics	Fine Aggregate	Coarse Aggregate	
		NA	RA
Dry-compacted density (kg/m^3^)	1614	1534	1267
Water absorption (%)	0.95	0.54	7.94
10%-fine value (k.N)	-	154	123
Particle-density	2.65	2.70	2.34
Max.-aggregate size (mm)	4.75	12.5	12.5
Min.-particle size (mm)	-	4.75	4.75

**Table 3 materials-13-04112-t003:** Characteristics of glass fiber.

Property	Material Type	Filament-Diameter (mm)	Filament-Length (mm)	Youngs Modulus (GPa)	Tensile Strength (GPa)	Specific Gravity	Melting Point (°C)
Value	Alkali-resistant glass	0.014	6–12, 12–18 (mixed)	72	1.7	2.63	Above 900

**Table 4 materials-13-04112-t004:** Identity (ID) and composition of each mix.

Mix IDs	Cement (kg)	Silica Fume (kg)	Fine Aggregate (kg)	NA (kg)	RA (kg)	Water (kg)	Plasticizer (kg)	Glass Fiber (kg)	Slump (cm)
NC	475	0	650	1075	0	180	0.0	0	9.8
NC-SF5	451	24	650	1075	0	180	1.3	0	9.1
NC-SF10	428	48	650	1075	0	180	1.8	0	10.4
NC-GF	475	0	644	1069	0	180	2.5	13	8.7
NC-GF-SF5	451	24	644	1069	0	180	3.1	13	8.2
NC-GF-SF10	428	48	644	1069	0	180	3.4	13	9.6
RC	475	0	650	0	994	180	0.0	0	9.5
RC-SF5	451	24	650	0	994	180	1.3	0	10.4
RC-SF10	428	48	650	0	994	180	1.8	0	9.2
RC-GF	475	0	644	0	988	180	2.5	13	8.7
RC-GF-SF5	451	24	644	0	988	180	3.1	13	9.5
RC-GF-SF10	428	48	644	0	988	180	3.4	13	10.6

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
