# Peer review of "Enhancing the Hardened Properties of Recycled Concrete (RC) through Synergistic Incorporation of Fiber Reinforcement and Silica Fume"

_materials, 2020, doi:10.3390/ma13184112_

Round 1

Reviewer 1 Report

The reviewer considers that this paper describes valuable test results for recycled aggregate concrete containing silica fume and glass fiber. The reviewer requests revises about following items.
1. Line 116: "2" should be superscripted.
2. Lines 137, 140, 145: "3" should be superscripted.
3. Line 139: "Table 3" -> "Table 4".
4. Section 2.2: The actual measured slump should be described.
5. Line 166: "56" -> "56-days".
6. Figure 4 and Figure 6: Reference number should be indicated. (References from #67 to #70 have not been mentioned in the body text.)
7. Line 218: "4.52%, 7.58%, 6.69%" should be "5%, 8%, 7%".
8. Line 288: "2" should be superscripted.
9. References: #11 and #18 have been listed doubly.
10. References: #13 and #72 have been listed doubly.

Reviewer 2 Report

Whilst it correctly claimed inline 75 onwards that fibres do not only increase mechanical perforamnce but do not give a correct reason for controlling degradation for the likes of freeze-thaw or other crack-type of damages.  This is because of the tesile toughness introduced by the presence of the fibre. Specifically this is due to the area seen beneath the tesion stress-strain curve manifest in either direct tension tests or implied energy absorption improvement in flexural beam bending tests. 

However, this paper presents no such tests so it is considered the claims made of perforamce improvement have not been sufficiently substantiated to justify publication at this time.  In particular, the fibre-reinforcement is expected to make a substantial improvement in this regard, but not the silica fume. The latter rather makes a denser mix, and therefore superior bond adhesion of the cement paste which, as the results show, give a marginal increase in compression and craking strength.  Additional margins of strength increase are achieved by the addition of the fibers, but this belies the true performance enhancement.  The ductility increase in this reviewer's experience would probably be in the order of 1000 percent with fibres.  The tensile behavior essentially is transformed from tension-brittle to elasto-plastic highly ductile.  It is this attribute that markedly modifies the performance and abates cracking significantly.  

To publish this paper and substantiate the claims that are made at present, needs the addition of tension behavior.  Without such substantiation, this paper should not be published, as the evidence is too thin based on the tests presented; there is also a lack of statistical robustness in the results [too few samples] to justify the claims as well. 

If the authors come back with more physical test results, showing enhanced tensile stress-strain performance, then I would be glad to rereview the paper.  Otherwise the paper should be summarily rejected based on its present form.

Round 2

Reviewer 2 Report

The addition of the flexural toughness is a very important addition that redeems the paper from extinction. I am happy to say that in my present opinion the paper may now be published